# Do generational diversity and perceived similarity improve team functioning in rural Chinese hospitals? A cross-sectional survey study

Hujie Wang  ,[1] Martina Buljac-Samardžić,[1] Jeroen van Wijngaarden,[1] Joris van de Klundert[1,2]

[1]Erasmus School of Health Policy & Management, Erasmus University Rotterdam, Rotterdam, The Netherlands
[2]Universidad Adolfo Ibáñez, Santiago, Chile

**Correspondence to**
Dr Hujie Wang;
wang@eshpm.eur.nl

## ABSTRACT

**Objectives** Generational diversity, increasingly prominent in the composition of the healthcare workforce in rapidly developing countries, has received much attention in practice and research recently. While research has revealed various positive and negative impacts of generational diversity on team functioning, the understanding of the mechanism explaining how generational diversity influences team functioning is still limited. This study in rural Chinese hospitals examines the relationship between (surface-level) generational diversity and (deep-level) perceived similarity and investigates how they influence three teamwork behaviours that importantly determine quality of care, namely speaking up, silence and knowledge sharing.

**Design** We adopted a quantitative research design and conducted an online survey to investigate the relationship among generational diversity, perceived similarity, speaking up, silence and knowledge sharing. Multilevel mediation modelling was used to test the hypotheses.

**Setting** The study was conducted in four rural Chinese hospitals.

**Participants** 841 healthcare professionals, including doctors, nurses and other healthcare professionals, were included in the study.

**Primary and secondary outcome measures** Generational diversity was measured by calculating the average of individuals' Blau's indices regarding all the generations for each team. Perceived similarity, speaking up, silence and knowledge sharing were measured by validated questionnaires from literature.

**Results** Perceived similarity is positively related to the three teamwork behaviours, that is, speaking up ($\beta=0.56$, $p<0.01$), silence ($\beta=0.39$, $p<0.01$) and knowledge sharing ($\beta=0.54$, $p<0.01$), while generational diversity is not (speaking up: $\beta=0.08$, $p>0.05$; silence: $\beta=0.44$, $p>0.05$; knowledge sharing: $\beta=0.09$, $p>0.05$). As the relationship between generational diversity and perceived similarity is non-significant ($\beta=0.07$, $p>0.05$), perceived similarity does not mediate the relationship between generational diversity and teamwork behaviour.

**Conclusion** The findings suggest that increases in generational diversity that result from healthcare workforce strengthening may not impact team behaviours and performance. However, if healthcare workforce strengthening would reduce the perceived similarity

## STRENGTHS AND LIMITATIONS OF THIS STUDY

⇒ Multilevel analysis was used to provide more accurate insights into the relationships among the variables by eliminating the cluster effects of the data.
⇒ Cross-sectional design does not allow to claim causal relationships.
⇒ Using one method to collect data from the same respondents may produce common method bias and common resource bias.

in teams, explicit management efforts to mitigate the negative impact on team behaviour and care provision are called for.

## INTRODUCTION

Sustainable Development Goal 3 (SDG3) aims at good health and well-being for all and at universal health coverage (UHC).[1] However, many developing countries experience barriers to high-quality care delivery, a key component of UHC, due to healthcare workforce shortages. These shortages are particularly severe in rural and remote areas.[2–4] Improving healthcare workforce management in developing countries, including recruitment, development and retention, can be an effective strategy towards achieving UHC.[1] Attracting and retaining a highly qualified workforce often requires embracing diversity in the workforce and healthcare teams. This diversity includes variation in generations, professional background and cultural or geographical origins.

Workforce diversity can both positively (eg, better health outcomes) and negatively (eg, increased conflict) impact healthcare team performance.[5 6] However, evidence is predominantly from developed countries,[5 6] leaving the role of workforce diversity in the performance of rural healthcare teams in

BMJ Group

developing countries, where UHC is most challenging, under-researched.

This study focuses on diversity of healthcare teams, particularly generational diversity, as a potential pathway to address workforce shortages and provide high-quality care.[7] The evidence regarding the relationship between generational diversity and team functioning is limited, inconclusive and primarily from developed countries.[8 9] For example, generational diversity may promote innovative team behaviour but also create communication barriers.[10 11] We aim to expand the evidence base by examining the relationship between diversity, particularly generational diversity, and healthcare team functioning in rural Chinese hospitals.

In recent decades, many developing countries such as China have experienced rapid and profound economic development and social changes, leading to larger generational differences compared with developed countries with more modest development rates. For instance, since initiating social reforms and opening up in 1978, China has achieved an average yearly gross domestic product (GDP) growth rate of 9.1%. This rate is nearly three-fold the 3.1% yearly growth of the global GDP during the same period.[12] Such growth can amplify the differences in behaviours and values between generations[13]; a generation is defined as a group of people sharing birth years in a certain period and raised in a similar social and development stage.[14 15] These differences in behaviours and values may subsequently influence team functioning and performance.

Rural Chinese hospitals have traditionally employed a less diverse workforce, mostly drawn from the local population, and have struggled with attracting highly educated healthcare professionals.[16] For these rural hospitals, which serve nearly half a billion rural Chinese citizens,[17] recruiting young, highly educated and non-local professionals increases (generational) diversity and supports delivering high-quality care as part of UHC.[1] To the best of our knowledge, existing evidence on the challenges posed by (generational) diversity in rural healthcare teams, both in China and elsewhere, is scarce and predominantly qualitative.[2 16 18] This study presents a first quantitative study into the relationship between diversity and teamwork behaviour in rural Chinese hospital teams.

Generational diversity is based on demographic characteristics and is therefore considered a form of surface-level diversity.[19 20] It may lead to perceived (dis)similarities between generations regarding communication styles and work attitudes.[21 22] These perceived (dis)similarities are forms of deep-level diversity, as they involve underlying attributes such as beliefs, attitudes and values.[20] Jansen and Searle's review on team diversity recommends researching surface-level and deep-level diversity simultaneously.[23]

Empirical evidence and established theories support a relationship between perceived similarity and team functioning. Shemla *et al*'s review reports that perceived similarity is positively related to team members' supportive behaviour (ie, social and task exchange) and team commitment while reducing employees' intention to resign.[24] Jansen and Searle's review also highlights the benefit of perceived similarity for team performance, team efficiency and job satisfaction.[23] The similarity attraction theory[25] suggests that similar team members are more likely to understand each other's thoughts and behaviours, increasing interaction.[20 26] In addition, social identity theory[27] and self-categorisation theory[28] propose that team members may categorise similar team members as 'ingroups' with whom they prefer to interact, while less similar team members are viewed as 'outgroups', resulting in less interaction.

As perceived similarity may be directly associated with communication and interaction, we operationalise team functioning through three individual behaviours essential to healthcare delivery: speaking up, silence and knowledge sharing. Silence and speaking up refer to the extent individuals voice work-related issues. Research has demonstrated the positive effect of speaking up[29 30] and the negative effect of silence[31] on team functioning in healthcare. Over the past years, knowledge sharing has also gained attention in healthcare.[32–34] Team members might still share knowledge to promote team functioning even when the team climate does not support speaking up or when it is not seen as appropriate.

If generational diversity translates into perceived similarity, which in turn impacts speaking up, silence and knowledge sharing, perceived similarity acts as a mediating mechanism through which generational diversity impacts the three teamwork behaviours. This study investigates how generational diversity and perceived similarity influence speaking up, silence and knowledge sharing in rural Chinese hospitals.

## Hypotheses
### Relationship between generational diversity and perceived similarity

Generational diversity occurs when a team includes members from at least two generations. Commonly considered generations in scientific literature include baby boomer (born between 1946 and 1964), Generation X (born between 1965 and 1981) and Generation Y (born between 1982 and 2000).[35] This categorisation is rooted in Western context and evidence, and it may not have external validity in rural China which has developed quite differently. In this study, we follow the generational categorisation commonly used in China-related literature which distinguishes the decades '1960s, 1970s, …' to define corresponding generations.[13 36–38] Each of these generations possesses distinct characteristics that are relevant to team functioning. For example, in China, people born in the 1970s are more collectivist and pragmatic, while those born in the 1980s are more individualistic and confident.[13 37]

Perceived similarity refers to one's perception of the similarity between people[39] and is related to deep-level attributes (ie, underlying elements such as values,

attitudes and beliefs).[20] (Surface-level) generational diversity may underpin perceived similarity through generational differences in values, attitudes and beliefs.[8 40] In a generationally diverse team, people from different generations may hold distinct values, attitudes and beliefs and will, therefore, perceive those from other generations as dissimilar to themselves. For the same reason, team members may be more likely to consider other team members from the same generation to be as similar to themselves. Therefore, we propose:

*Hypothesis 1*: Generational diversity is negatively related to perceived similarity.

### Relationship between perceived similarity and teamwork behaviours

Speaking up, also known as voice behaviour, is defined as expressing one's opinions about any work-related matters.[41 42] Employee silence refers to 'withholding ideas, information and opinions' relevant to the improvements in workplace.[43] Speaking up and silence are not necessarily opposites and can coexist. Individuals who speak up on certain issues may still intentionally withhold their ideas and keep silent about other issues at the same time.[43 44] Knoll and Redman even found that some types of speaking up and silence behaviours (ie, promotive speaking up and cooperative silence) are positively correlated.[45]

Sometimes, team members may not be able or willing to point out work-related issues but may still practise knowledge sharing. Knowledge sharing refers to people sharing their work-related knowledge with team members to help improve team effectiveness.[46 47]

Antecedents of speaking up, silence and knowledge sharing are identified by previous reviews and studies. For example, some reviews summarise the factors influencing speaking up and silence behaviour (eg, motivation, contextual factors, individual factors and organisational factors).[30 48–50] Wang and Noe have studied the factors influencing knowledge sharing and provide evidence that diversity in demographic features (eg, gender, education and marital status) is associated with knowledge sharing in non-healthcare settings.[51] Workplace ostracism has been shown to be negatively related to speaking up[52 53] and knowledge sharing[54 55] but to be positively associated with silence[53 56] and knowledge hiding.[57 58]

The similarity attraction theory suggests that people are attracted to those similar to themselves but may ostracise those who are dissimilar.[25] According to this theory, individuals more easily build good interpersonal relations with similar ones, understand their thoughts and behaviours better and are subsequently more likely to interact with them.[20 26] In line with this reasoning, team members would be more likely to speak up and share knowledge within the team and less likely to withhold their ideas if they perceive themselves to be more similar to each other. Conversely, team members would be more reluctant to express their ideas and share their knowledge and more prone to keeping silent with team members perceived as dissimilar. Therefore, we propose:

*Hypothesis 2a*: Perceived similarity is positively related to speaking up.

*Hypothesis 2b*: Perceived similarity is negatively related to silence.

*Hypothesis 2c*: Perceived similarity is positively related to knowledge sharing.

### The mediating role of perceived similarity

The similarity attraction theory posits that individuals' actual similarity (eg, demographic characteristics) leads to perceived similarity and consequently influences behaviour.[25] As generational diversity creates dissimilarity, individuals may perceive team members from other generations as dissimilar, causing them to be less likely to speak up, more likely to remain silent and less likely to share knowledge. Similarly, low generational diversity within teams, as occurs when there are few generational differences between team members, would imply high perceived similarity, promoting speaking up and knowledge sharing and reducing silence. We accordingly propose:

*Hypothesis 3*: Perceived similarity mediates the relationship between generational diversity and (a) speaking up, (b) silence and (c) knowledge sharing.

The theoretical model of this study is shown in figure 1.

## METHODS
### Sample and procedure

Following our research aims, we approached rural hospitals in China. Specifically, we approached seven

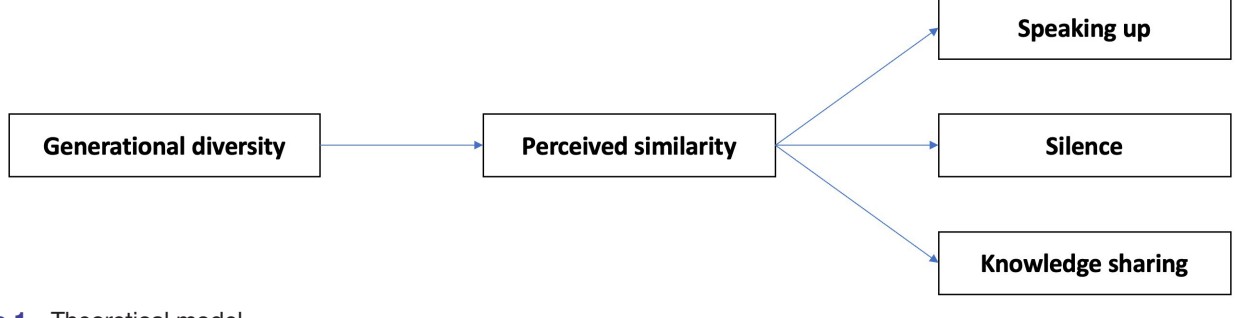

**Figure 1** Theoretical model.

county-level hospitals, with the help of the Health Human Resources Development Centre of the National Health Commission of China and the County Health Media. The former is responsible for promoting the human resource management for hospitals of all levels in China, while the latter is a media company focusing on China's county-level hospitals. Four out of the seven county-level hospitals accepted our invitation to participate in the study. These four hospitals are located in four different provincial administrative regions across China. Together, these four hospitals employ 3500 employees, including approximately 1000 doctors and 1700 nurses.

This study was conducted anonymously through an online cross-sectional survey, which was disseminated via the widely used Chinese survey platform 'Wen Juan Xing' in October 2022. The hospital presidents of the four participating hospitals explained the research aims to the team leaders and asked them to share the survey link with team members. Participants were team members, including doctors, nurses and other healthcare professionals, working in monodisciplinary teams at the four county-level hospitals. Team leaders were included only for the measurement of generational diversity, but not for the other constructs. Each respondent was restricted by the survey platform to submit only one response per item.

## Measures

### Generational diversity

Following existing literature on the definition of generations in China, we specified the generations into five categories: born between 1960 and 1969, born between 1970 and 1979, born between 1980 and 1989, born between 1990 and 1999, and born after 2000.[13 36–38] Team members and team leaders were asked to report the number of people for each category on their teams. The generational diversity perceived by each participant was calculated using Blau's index[59]: $1 - \sum_{i=1}^{n} p_i^2$, where $p_i$ is the perceived proportion of team members in the $i^{th}$ category (generation), and $n$ is the perceived number of categories (generations) in the team. Blau's index equals 0 if all team members are from the same generation and increases as the members are divided across more generations. As generational diversity is a team-level construct, we calculated the mean of the individual Blau's indices of team members and team leaders for each team to express perceived generational diversity at the team level.

### Perceived similarity

We measured the perceived similarity between team members using Williams *et al*'s six-item measure,[20] which was adapted from Liden *et al*.[60] This scale is a 7-point Likert scale, ranging from 1 (strongly disagree) to 7 (strongly agree). The Cronbach's α in this study was 0.97.

### Speaking up

Speaking up was measured by a six-item measure, developed by Van Dyne and LePine[61] and adapted by Morrison *et al*.[62] This 7-point Likert scale ranged from 1 (strongly disagree) to 7 (strongly agree). The Cronbach's α in this study was 0.94.

### Silence

A five-item measure developed by Detert and Edmonson[63] and adapted by Guenter *et al*[64] and Mignonac *et al*[65] was used to test employee silence. The items were rated on a 7-point Likert scale ranging from 1 (strongly disagree) to 7 (strongly agree). The Cronbach's α in this study was 0.94.

### Knowledge sharing

We assessed knowledge sharing with Pittino *et al*'s five-item measure,[66] as adapted from Bartol *et al*.[67] A 7-point Likert scale was used, ranging from 1 (strongly disagree) to 7 (strongly agree). The Cronbach's α in this study was 0.99.

### Control variables

From a wide set of literature-based control variables, gender and team tenure were significantly associated with one of the dependent variables in preliminary analyses and were therefore included.[20 61 62 64 67 68] Furthermore, we also controlled for team size at the team level as it has been found to influence speaking up and other interaction-related variables within a team in previous research.[69 70]

The measures for perceived similarity, speaking up, silence and knowledge sharing are presented in online supplemental appendix 1. All measures were translated from English to Chinese using the standard translation/back-translation technique.[71] The average scores of the items for each scale were used to form the individual measurements of perceived similarity, speaking up, silence and knowledge sharing for each participant.

## Analysis

All the analyses were conducted via SPSS V.29 and AMOS V.28. We conducted descriptive and correlation analyses via SPSS and confirmatory factor analysis (CFA) via AMOS to provide an overview of the data. The CFA shows good factor loadings for perceived similarity, speaking up, silence and knowledge sharing, ranging from 0.72 to 0.99. The model fit indices of the CFA suggest that the four-factor model (ie, a model with perceived similarity, speaking up, silence and knowledge sharing being separate factors) presents an acceptable model fit ($\chi^2(203)=1796.92$, Comparative Fit Index (CFI)=0.94, Tucker-Lewis Index (TLI)=0.94, Root Mean Square Error of Approximation (RMSEA)=0.10, Standardised Root Mean Square Residual (SRMR)=0.05) and fits significantly better than the three-factor model (ie, combining speaking up and silence into one factor; $\chi^2(206)=5491.54$, p<0.01, CFI=0.81, TLI=0.79, RMSEA=0.18, SRMR=0.14), two-factor model (ie, combining speaking up, silence and knowledge sharing; $\chi^2(208)=8163.26$, p<0.01, CFI=0.71, TLI=0.68, RMSEA=0.21, SRMR=0.18) and one-factor model (ie, combining all the four as one

factor; $\chi^2(209)=12\,430.70$, p<0.01, CFI=0.56, TLI=0.51, RMSEA=0.26, SRMR=0.21).

Multilevel analysis was used to analyse the data as respondents were nested within teams, and teams were further nested within hospitals, which caused dependencies between observations. The standard mixed linear models in SPSS showed that there were significant between-team variances for the three dependent variables (ie, speaking up, silence and knowledge sharing), indicating the necessity of conducting a multilevel analysis. In addition, between-hospital variances were not significant. Therefore, a two-level mediation analysis was performed; within-group (level 1) represented the individual level, while between-group (level 2) represented the team level. We chose the random-effects model for the multilevel mediation analysis as it is expected that the intercepts and slopes would vary across higher level groups (ie, teams).[72]

We used the MLmed macro in SPSS to conduct the multilevel mediation analysis.[73] In the interface of the MLmed macro, three different random-effects models were built with 'speaking up', 'silence' and 'knowledge sharing' (level 1) as dependent variables. 'Generational diversity' (level 2) served as the independent variable, while 'perceived similarity' (level 1) served as a mediator. These three models were 2-1-1 multilevel mediation models in which the independent variable was a team-level (level 2) variable, and the other variables were at the individual level (level 1). The control variables gender and team tenure were level 1 covariates, and team size was a level 2 covariate.

As generational diversity is a level 2 variable, the random-effects parameters only account for the intercepts of the mediator and dependent variables and the slopes regarding the relationships between perceived similarity and the three behavioural variables (ie, speaking up, silence and knowledge sharing) as these four variables are all at level 1.

Additional multilevel mediation analyses replacing generational diversity by the diversity in the composition of local and non-local healthcare professionals were conducted to provide more insights for future research. The reporting of this study follows the Checklist for Reporting of Survey Studies.[74]

### Patient and public involvement

Patients or the public were not involved in the design and reporting of our research. Nevertheless, the Health Human Resources Development Centre of the National Health Commission of China and the hospital management of the four participating hospitals play a supporting role in conducting the research, aiming for quality and teamwork improvement in all rural hospitals. Therefore, we will disseminate our findings with the participating hospitals to improve team functioning and subsequently the quality of care.

**Table 1** Demographic characteristics

|  | People, n (%) |
|---|---|
| Gender | |
| Male | 165 (19.62) |
| Female | 652 (77.53) |
| Prefer not to say | 24 (2.85) |
| Age* | |
| ≤30 | 414 (49.23) |
| 31–40 | 287 (34.13) |
| 41–50 | 103 (12.25) |
| ≥51 | 32 (3.80) |
| Profession | |
| Doctors | 325 (38.64) |
| Nurses | 479 (56.96) |
| Other healthcare professionals | 37 (4.40) |
| Local or non-local | |
| Local | 731 (86.92) |
| Non-local | 110 (13.08) |
| Education background | |
| Master | 18 (2.14) |
| Bachelor | 593 (70.51) |
| Lower than Bachelor | 230 (27.35) |
| Professional title | |
| Senior | 17 (2.02) |
| Deputy senior | 55 (6.54) |
| Intermediate | 215 (25.56) |
| Junior | 554 (65.87) |

*There are missing values regarding age, so the sum of the number of people per age group is smaller than the total number of respondents.

## RESULTS

### Descriptive analysis

A total of 841 valid questionnaires were received. The average age of the respondents is 32.10 years (median: 31.00; SD: 8.08). The average team tenure is 6.71 years (median: 5.00; SD: 6.37). The percentage of doctors among the respondents (38.64%) is similar to that in all Chinese hospitals (38.04%), while the percentage of nurses (56.96%) is higher than the national data (44.81%).[75] The respondents' demographic characteristics are shown in table 1.

The correlation analyses (table 2) show significant strong correlations between perceived similarity on the one hand and speaking up (r=0.65, p<0.01) and knowledge sharing (r=0.67, p<0.01) on the other hand. The analysis finds a small to moderate correlation between perceived similarity and silence (r=0.26, p<0.01). Significant strong correlation is also shown between speaking up and knowledge sharing (r=0.75, p<0.01), while the positive correlation between speaking up and silence is

**Table 2** The correlation matrix of all variables

|  | 1 | 2 | 3 | 4 | 5 | 6 | 7 | 8 |
|---|---|---|---|---|---|---|---|---|
| 1. Gender (1=female) | 1.00 | | | | | | | |
| 2. Team tenure | −0.19* | 1.00 | | | | | | |
| 3. Team size | 0.02 | 0.10* | 1.00 | | | | | |
| 4. Generational diversity† | −0.14* | 0.23* | 0.22* | 1.00 | | | | |
| 5. Perceived similarity | 0.07 | 0.00 | −0.03 | −0.00 | 1.00 | | | |
| 6. Speaking up | 0.02 | 0.04 | −0.06 | 0.01 | 0.65* | 1.00 | | |
| 7. Silence | −0.02 | 0.06 | −0.01 | 0.04 | 0.26* | 0.33* | 1.00 | |
| 8. Knowledge sharing | 0.12* | −0.01 | −0.03 | 0.00 | 0.67* | 0.75* | 0.15* | 1.00 |

*P<0.01.
†The generational diversity is a team-level construct, and thus a single value is assigned to all the individuals from the same team within the range of 0 and 0.8.

moderate (r=0.33, p<0.01). Generational diversity at team level is not significantly correlated to either perceived similarity or individual behaviours. The full collinearity test shows that the values of the variance inflation factor for all the control variables, independent variable and mediator range from 1.01 to 1.12, indicating no multicollinearity issues in this study.[76]

### Multilevel mediation analysis

The three 2-1-1 multilevel mediation models are shown in table 3.

Table 3 shows that there is no significant relationship between generational diversity and perceived similarity (β=−0.07, p>0.05). Therefore, hypothesis 1 is rejected, which also violates the first requirement for a mediating relationship.

The results also show that perceived similarity is positively related to all the three behavioural variables at both the within-group level (speaking up: β=0.56, p<0.01; silence: β=0.39, p<0.01; knowledge sharing: β=0.54, p<0.01) and between-group level (speaking up: β=0.69, p<0.01; silence: β=0.38, p<0.01; knowledge sharing: β=0.60, p<0.01). These findings indicate that people will be more likely to speak up, keep silent and share knowledge when they perceive themselves to be more similar to team members. As a result, these results support hypotheses 2a and 2c but reject hypothesis 2b, which specifies a negative relationship between perceived similarity and silence.

The random-effects parameters show that the relationships between perceived similarity and speaking up (0.12,

**Table 3** Multilevel mediation analyses

|  | Perceived similarity | Speaking up | Silence | Knowledge sharing |
|---|---|---|---|---|
| ***Fixed effects*** | | | | |
| **Within-group (level 1)** | | | | |
| Intercept | 5.46** | 1.75** | 1.65* | 2.39** |
| **Perceived similarity** | – | 0.56** | 0.39** | 0.54** |
| Gender | 0.23 | 0.04 | −0.31 | 0.30** |
| Team tenure | −0.00 | 0.00 | 0.01 | −0.01 |
| **Between-group (level 2)** | | | | |
| **Generational diversity** | 0.07 | 0.08 | 0.44 | 0.09 |
| **Perceived similarity** | – | 0.69** | 0.38** | 0.60** |
| Gender | 0.27 | −0.15 | 0.11 | 0.14 |
| Team tenure | 0.01 | 0.01 | 0.02 | 0.00 |
| Team size | −0.00 | −0.00 | −0.00 | −0.00 |
| **Indirect effect (mediation)** | – | 0.05 | 0.03 | 0.04 |
| ***Random effects (variance)*** | | | | |
| Intercept | 0.19** | 0.03 | 0.41** | 0.04 |
| Slope | – | 0.12** | 0.04 | 0.11** |

*P<0.05; **p<0.01.

p<0.01) and between perceived similarity and knowledge sharing (0.11, p<0.01) significantly vary across groups, while the relationship between perceived similarity and silence (0.04, p>0.05) does not significantly change between groups. These findings illustrate that within the teams with the same extent of perceived similarity, team members in some of these teams are more willing to speak up and share knowledge than those in some other teams, but the extent of keeping silent does not vary across teams. Furthermore, the intercepts of perceived similarity and silence vary across groups, while those of speaking up and knowledge sharing do not. These results indicate that team members from different teams have different baseline degrees of the perceptions towards other team members and keeping silent. However, the baseline extents of speaking up and sharing knowledge remain constant across teams.

As generational diversity is a level 2 variable, only between-group direct effects (ie, the relationships between generational diversity and the three behavioural variables) and mediated effects (ie, indirect effects) are evaluated. As shown in table 3, there are neither significant direct effects of generational diversity on the three behavioural variables (speaking up: β=0.08, p>0.05; silence: β=0.44, p>0.05; knowledge sharing: β=0.09, p>0.05) nor significant mediated effects via perceived similarity (speaking up: β=0.05, p>0.05; silence: β=0.03, p>0.05; knowledge sharing: β=0.04, p>0.05). Therefore, hypothesis 3 is not supported.

## DISCUSSION

This study investigates the relationships among generational diversity, perceived similarity, speaking up, silence and knowledge sharing in rural Chinese hospitals. In contrast to the theory-informed hypotheses formulated, this study finds no significant relationships between generational diversity and the other four variables (ie, perceived similarity, speaking up, silence and knowledge sharing) and therefore appears to exclude significant direct and mediated effects of generational diversity on these variables.

Generational diversity is defined based on age groups and is therefore considered surface-level attribute.[19 20] Literature shows that surface-level similarity does not necessarily align with perceived similarity.[20 77] We attempted to establish the connection between generational diversity and perceived similarity via intergenerational differences as the latter two are related to deep-level attributes (eg, values, attitudes and beliefs) but found non-significant relationship. Other determinants of perceived similarity exist as well, which may influence perceived similarity more strongly. For instance, Ott-Holland et al demonstrate that gender plays a role in perceived similarity, and females perceive themselves to be more similar to others than do males.[78] Wang et al found that local origin may impact perceived similarity.[16] People raised in the same area share the local culture, norms and social background

and therefore are more likely to form similar values and beliefs. Of the respondents included in this study, 86.85% were born and raised in the areas where their hospitals are located. An additional analysis, however, reveals that the composition of local and non-local employees is not significantly related to perceived similarity, nor to individual behaviours (online supplemental appendix 2). Therefore, more research is needed to understand the determinants of perceived similarity.

Our findings show that perceived similarity is positively associated with speaking up and knowledge sharing at both individual and team levels. These findings confirm the similarity attraction theory, which posits such interactive behaviours to be associated with similarity.[25] The social identity theory[27] and the self-categorisation theory[28] provide further theoretical support for this finding. Similar people will categorise themselves as 'ingroups', enhance their self-image and then be more willing to interact with each other. Conversely, individuals perceive dissimilar ones as 'outgroups' and amplify the differences between 'ingroups' and 'outgroups', which may form intergroup conflicts and ultimately reduce the frequency of intergroup interactive behaviours. The evidence found and the theoretical support thus indicate that increasing the recruitment of healthcare workforce in rural Chinese hospitals (and other rural settings) to strengthen the quality of care and UHC may negatively influence speaking up and knowledge sharing and impede teamwork if it causes increased perceived dissimilarity.

Counter to our hypothesis, perceived similarity is positively related to silence at both individual and team levels. The social identity theory[27] and the self-categorisation theory[28] may provide explanations for this finding. People consider similar team members to be part of their 'ingroups', which will stimulate personal relationships. These personal relationships may subsequently lead them to refrain from commenting openly on the functioning of these 'ingroup' members. As this study was conducted in rural China, the Chinese cultural values of 'saving face' and 'harmony' to promote team functioning may further amplify this mechanism.[79] This cultural perspective sheds a novel light on the common, perhaps Western, view that silence is harmful to team functioning in healthcare.[31] Furthermore, team members may be less likely to remain silent with dissimilar team members ('outgroups'), as perceived dissimilarity is often accompanied by dissatisfaction with others.[80 81]

In addition to the positive association between perceived similarity and both speaking up and silence, we somewhat surprisingly find a positive correlation between the latter two. This correlation may be explained by distinguishing various subtypes of speaking up and silence. Scientific literature has identified acquiescent speaking up/silence (expressing/withholding ideas based on resignation and low self-efficacy), defensive speaking up/silence (expressing/withholding ideas based on self-protection and fear) and prosocial speaking up/silence (expressing/withholding ideas based on

benefiting the organisation and cooperation).[43] These different forms of speaking up and silence can explain why team members may or may not express constructive suggestions; for example, speaking up with the goal of benefiting the team while keeping silent to protect themselves and their 'ingroups' on other occasions. Our findings from the rural Chinese context thus confirm that speaking up and silence are distinct constructs rather than a pair of opposite behaviours.[43 44] It is worthwhile to investigate subtypes of speaking up and silence, and their relationship with perceived similarity in future research on team functioning and performance.

In a nutshell, this study has provided insights into the influence diversity has on team functioning in rural Chinese hospitals. The findings suggest that rural hospitals' efforts to recruit a more generationally diverse healthcare workforce in recent years are not likely to have directly impacted team functioning or the perceived similarity. At the same time, our results reveal that perceived similarity is likely to enhance teamwork behaviour. Hence, the effects of the necessary workforce strengthening efforts on perceived diversity deserve continued management attention and support. The benefits of similarity need to be balanced with the value of (generational) diversity in team functioning in rural Chinese hospitals, as also confirmed by evidence from Western countries.[5 6] There is qualitative evidence that older doctors appreciate the knowledge and energy brought by the younger generations.[16] Future qualitative and quantitative studies are needed to further explore the role of perceived similarity in team functioning in rural Chinese hospitals and the mechanisms by which hypothesised antecedents such as generational diversity impact perceived similarity. These mechanisms and antecedents may be context specific, and further research on the role of diversity in strengthening the rural health workforce in pursuit of SDG3 is therefore also called for.

### Practical implications

Our study shows that teamwork behaviour is associated with perceived similarity yet less so with generational diversity. This is relevant because the teamwork behaviours studied (ie, speaking up, silence and knowledge sharing) are associated with team performance and quality of care. Thus, rural Chinese hospital management may leverage the benefits of perceived similarity for team functioning to improve the quality of care and UHC. At the same time, our findings indicate that increases in generational diversity, which may result from ongoing workforce strengthening efforts, will not negatively impact the studied teamwork behaviours in support of healthcare quality and achieving UHC. However, when recruitment efforts for workforce strengthening lead to greater perceived dissimilarity in teams, these dissimilarities need to be managed, for instance, through team training to avoid negative impacts on team functioning and, possibly, quality of care.

### Strengths and limitations

This study provides insights into the relationship between generational diversity, perceived similarity and team functioning in rural Chinese hospitals based on a large survey study. Four Chinese county-level hospitals were included in the data collection. Although this resulted in a sufficient number of respondents, the selection of solely county-level hospitals may lead to a selection bias and may not depict a full picture of all rural Chinese hospitals. Moreover, as the number of people who received the survey link is unknown, the response rate is unclear.

Nevertheless, the four participating hospitals represent considerable geographical diversity with two of them being from distinct northern Chinese regions, and the other two from distinct southern Chinese regions. The sample size and variation of demographic characteristics and backgrounds are a strength of this study and ensure that the respondent sample represents the workforce in rural county-level hospitals and differs from national and provincial hospitals. This increases the generalisability to other rural county-level hospitals and at the same time limits the generalisability to other types of hospitals. This survey was constructed on the basis of validated measures, which strengthens the survey design. Nevertheless, all the variables were measured by the same respondents and via the same data collection method (survey), which may produce common source bias and common method bias. Lastly, the interpretation of our findings should take into account that the cross-sectional design does not allow claims about causality of the significant relationships between perceived similarity and the three teamwork behaviours studied.

### CONCLUSION

Perceived similarity is positively related to speaking up, silence and knowledge sharing at both individual and team levels. Generational diversity has no significant relationships with perceived similarity nor with these three teamwork behaviours. Thus, if healthcare workforce strengthening increases generational diversity, this may not impact team behaviour and performance. However, explicit management efforts to mitigate the negative impact on team behaviour and care provision are called for if the healthcare workforce strengthening required for improving quality of care and UHC reduces perceived similarity in teams. More research is needed to explore the antecedents of (deep-level) perceived similarity and the interaction between perceived similarity and various forms of speaking up and silence.

**Acknowledgements** We thank Yun Zhang and Jing Zhou from the Health Human Resources Development Centre of the National Health Commission of China, and Yan'an Wang and Xiaoxiao Wan from the County Health Media for helping connect Chinese hospitals. We are also grateful for Wenxing Wang's participation in the translation/back-translation before data collection. Most importantly, we show our highest gratitude to the presidents and all the respondents from the four rural Chinese hospitals for their contributions to this study.

**Contributors** HW, MB-S, JVW and JvdK designed the study. HW collected the data. HW and MB-S analysed the data. HW drafted the manuscript. HW, MB-S, JVW and JvdK revised and approved the manuscript. HW is the guarantor.

**Funding** This study was supported by the China Scholarship Council (Grant No 201906160092).

**Disclaimer** The funder has no role in the study design, data collection, data analysis and writing the manuscript.

**Competing interests** None declared.

**Patient and public involvement** Patients and/or the public were involved in the design, or conduct, or reporting, or dissemination plans of this research. Refer to the Methods section for further details.

**Patient consent for publication** Not applicable.

**Ethics approval** This study involves human participants and was approved by the Research Ethics Review Committee of Erasmus School of Health Policy & Management, Erasmus University Rotterdam (No ETH2122-0807). Consent was obtained from each respondent before the data collection. Participants gave informed consent to participate in the study before taking part.

**Provenance and peer review** Not commissioned; externally peer reviewed.

**Data availability statement** Data are available upon reasonable request. The data that support the findings of this study are available from the corresponding author upon reasonable request.

**ORCID iD**
Hujie Wang http://orcid.org/0000-0003-1588-7874

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
