## [Reviewer comments · BMJ Open]

ARTICLE DETAILS

Title (Provisional)

Does (generational) diversity improve team functioning in rural Chinese hospitals? A cross-sectional survey study

Authors

Wang, Hujie; Buljac-Samardžić, Martina; Van Wijngaarden, Jeroen; van de Klundert, Joris

VERSION 1 - REVIEW

Reviewer	1
Name	Schmidt, Manuela
Affiliation	Jönköping University
Date	03-May-2024
COI	none

Dear Authors,

thank you for the opportunity to review your scholarship. You present an original idea and tough on an interesting topic, however I have several consideration that you can address or reflect upon.

Overall I suggest additional language editing throughout the manuscript.

Title: Why is healthcare workforce strengthening in the title? Its not really a concept that you use in your paper. I would skip it. I would also skip the brackets for "generational", because it is your core concept. No need of hinding it.

Abstract: Please remove the first meaning of your conclusions. It is just a repetition of your results.

Strength and limiations: I recommend that more thought should be given to this at the end of the paper.

Introduction: I recommend using a visual aid, a model that clearly indicates your expected correlations/mediation between your concepts.

Ethics: One of my major concerns is that you only obtained ethical approval in the Netherlands but not in China. Why is that?

Patient and public involvement: It might be a requirement from the journal but for me your comment raises more questions than it answered. Did you in any way attempt co-production on any level?

Results: have you performed a drop out analysis in any way as the data seems very screwed regarding age, profession and the professional title, the latter being of unclear meaning to me. How can only 18 people have a master when 325 had a doctor degree. And how can only 17 have a senior title? Why only nurses and doctors answered the survey?

Discussion: you are rather short in your discussion which I guess in a quantitative paper is common. I would recommend you tried to connect your results better to existing literature in the discussion. You also do not connect to your rural context in the discussion/conclusion that was so central to you to start with. Please connect back to that as well. So what do the results mean in the rural context? Why is it important with that specific context?

VERSION 1 - AUTHOR RESPONSE

Reviewer Report:

Reviewer 1 Comments to the Author:

Dear Authors, thank you for the opportunity to review your scholarship. You present an original idea and tough on an interesting topic, however I have several consideration that you can address or reflect upon. Overall I suggest additional language editing throughout the manuscript.

Response: *Thank you for your constructive comments. We have carefully checked and corrected all spelling and grammar errors throughout the manuscript.*

Title: Why is healthcare workforce strengthening in the title? Its not really a concept that you use in your paper. I would skip it. I would also skip the brackets for "generational", because it is your core concept. No need of hinding it.

Response: *We agree. The concept "healthcare workforce strengthening" connects our study with Sustainable Development Goal 3 rather than representing our core research contribution. We have deleted "healthcare workforce strengthening" from our title.*

Regarding the brackets around "generational", we believe it is important to emphasise that while generational diversity is a main topic of interest, the research also considers "perceived (dis)similarity", which is another form of diversity. General diversity is the starting point of our analysis, yet perceived dissimilarity appears to be more significantly associated with team behaviour. Therefore, we prefer to include "generational" in the title while using brackets to indicate that our study also researches diversity in a more general sense.

Abstract: Please remove the first meaning of your conclusions. It is just a repetition of your results.

Response: *Thank you for your suggestion. We have deleted the repetitive words from the conclusion.*

Strength and limitations: I recommend that more thought should be given to this at the end of the paper.

Response: *We have added the “Strengths and limitations of the study” section at the end of the manuscript and thereby expanded our previous paragraph on limitations.*

Introduction: I recommend using a visual aid, a model that clearly indicates your expected correlations/mediation between your concepts.

Response: *Thank you for your suggestion. We added a theoretical model at the end of the “Hypotheses” section to more explicitly show the relationships among the variables. Please see Figure 1.*

Ethics: One of my major concerns is that you only obtained ethical approval in the Netherlands but not in China. Why is that?

Response: *Thank you for raising this point. All the researchers are affiliated to Erasmus School of Health Policy & Management, Erasmus University Rotterdam and should comply with the ethical requirements of this university, which require us to obtain ethical approval from the ethics review committee of Erasmus School of Health Policy & Management for research conducted outside the European Union. On the basis of this approval and all consent forms and procedures provided, our counterparts in China indicated that no additional ethical approval was required from their part.*

Patient and public involvement: It might be a requirement from the journal but for me your comment raises more questions than it answered. Did you in any way attempt co-production on any level?

Response: *We included a “patient and public involvement” statement in the original manuscript. At that time, we thought this section was only regarding clinical research which involves patients, so we stated that patients and the public were not involved in any step of the research. However, we are now aware that our respondents (i.e. healthcare professionals), participating hospitals and governmental authorities should be seen as a part of the public. Therefore, we revised the statement to avoid confusion. We state that the public was still not involved in the design and reporting of the research, but that we appreciate the supporting role of the participating hospitals in conducting the research and will share the results with them after publication. The dissemination of our findings will help the participating hospitals improve team functioning and subsequently the quality of care. This would be beneficial for all patients visiting those hospitals. Please see the “Patient and public involvement” section on Page 9.*

Results: have you performed a drop out analysis in any way as the data seems very screwed regarding age, profession and the professional title, the latter being of unclear meaning to me. How can only 18 people have a master when 325 had a doctor degree. And how can only 17 have a senior title? Why only nurses and doctors answered the survey?

Response: *We were not able to perform a drop out analysis regarding the non-response. We do not know the characteristics of invited respondents who did not respond. In our limitations, we therefore state that the response rate was not able to be calculated. We also did not remove any data on the basis of the respondents' demographic profiles. We aimed to be inclusive in the respondent recruitment process and embrace the diversity and characteristics as they occur in the rural hospitals, as we also now explain in the strengths and limitations.*

Professional title refers to the level of expertise within a particular profession. For example, senior titles include chief physician (doctors), chief nurse (nurses) or chief technician (technicians). The level of expertise of healthcare professionals in rural Chinese hospitals is much lower than in the national and provincial hospitals, so it is difficult for the healthcare professionals in rural hospitals to be promoted to a senior or deputy senior professional title. That is why you found that there are only 17 people with a senior title and 55 with a deputy senior title.

Similarly, the educational background of healthcare professionals in rural hospitals is also much lower than their counterparts in higher-level hospitals. In our case, there are no doctoral degrees (i.e. PhD) and only 18 healthcare professionals with a master's degree, while others have a background of bachelor or even lower than bachelor. Indeed, it is possible to be considered a medical doctor without a master's degree in China.

Our study includes not only doctors and nurses but also other healthcare professionals. The reason for the small number of other healthcare professionals is that we only include teams that directly provide care for patients while excluding those such as pharmacy and laboratory. The majority of the healthcare professionals in the teams included are doctors and nurses. Therefore, our data reflect the real situation of healthcare teams in those rural hospitals.

Discussion: you are rather short in your discussion which I guess in a quantitative paper is common. I would recommend you tried to connect your results better to existing literature in the discussion. You also do not connect to your rural context in the discussion/conclusion that was so central to you to start with. Please connect back to that as well. So what do the results mean in the rural context? Why is it important with that specific context?

Response: *Thank you for your comments on the discussion section. We have expanded the discussion section by adding a paragraph to link our research back to rural context and try to generalise the findings to other rural areas in developing countries. The length of the discussion is indeed probably related to the research design. We build our hypotheses based on existing*

theories and knowledge. Therefore, we connected the results back to those theories, as seen in the discussion section. In addition, we also provided possible explanations for unexpected findings (e.g. the positive relationship between perceived similarity and silence) using other theories and literature, especially relevant to the rural context. As a quantitative study, we could not expand too much in the discussion as those possible explanations could be seen as speculations, which are not researched or supported by our study.